# ⏯ Vid-SME: Membership Inference Attacks against Large Video Understanding Models

**Qi Li    Runpeng Yu    Xinchao Wang***
National University of Singapore
{liqi, r.yu}@u.nus.edu   xinchao@nus.edu.sg

## Abstract

Multimodal large language models (MLLMs) demonstrate remarkable capabilities in handling complex multimodal tasks and are increasingly adopted in video under-standing applications. However, their rapid advancement raises serious data privacy concerns, particularly given the potential inclusion of sensitive video content, such as personal recordings and surveillance footage, in their training datasets. Deter-mining improperly used videos during training remains a critical and unresolved challenge. Despite considerable progress on membership inference attacks (MIAs) for text and image data in MLLMs, existing methods fail to generalize effectively to the video domain. These methods suffer from poor scalability as more frames are sampled and generally achieve negligible true positive rates at low false positive rates (TPR@Low FPR), mainly due to their failure to capture the inherent temporal variations of video frames and to account for model behavior differences as the number of frames varies. To address these challenges, we introduce Vid-SME (Video Sharma–Mittal Entropy), the first membership inference method tailored for video data used in video understanding LLMs (VULLMs). Vid-SME leverages the confidence of model output and integrates adaptive parameterization to compute Sharma–Mittal entropy (SME) for video inputs. By leveraging the SME difference between natural and temporally-reversed video frames, Vid-SME derives robust membership scores to determine whether a given video is part of the model's training set. Experiments on various self-trained and open-sourced VULLMs demonstrate the strong effectiveness of Vid-SME. Code is available here.

## 1   Introduction

Multimodal large language models (MLLMs) [1, 11, 26, 55] have received widespread attention from the AI community. By combining large language models (LLMs) with vision encoders, MLLMs gain the ability to perform a wide range of vision-language tasks [17, 64, 19, 21]. Recently, there has been growing interest in extending MLLMs to video understanding [63, 28, 30, 45], driven by their strong capabilities in processing visual information. However, the rapid development of video understanding LLMs (VULLMs) also raises critical concerns regarding data privacy leakage, as videos used for model training may contain sensitive content, such as personal recordings and surveillance footage, which could be memorized and unintentionally exposed by the models [5, 48, 61]. This highlights the severity of the problem, since early studies demonstrate that models' memorization of data can be maliciously exploited to conduct membership inference attacks (MIAs) [44, 4], where adversaries aim to determine whether a specific data sample was used during training. However, despite the booming development of VULLMs, efforts to address this issue significantly lags behind.

Recent studies have explored MIAs on LLMs and MLLMs [59, 43, 24, 50]. However, we observe that directly applying these methods to VULLMs results in extremely poor performance, and the

---

*Corresponding Author

performance often deteriorates as more frames are introduced. The underlying reason is that these methods adopt a static view of MIA, which is inconsistent with the temporal nature and complex inter-frame variations of video data. Moreover, they overlook the intricate relationship between MIAs and model performance variations across different frame conditions. Since MIAs fundamentally rely on identifying model memorization [44] and such memorization in VULLMs may vary with the frame conditions [34, 56], the model tends to provide substantially different inference signals to the adversary when processing different number of frames from the same video. Therefore, successful MIAs on VULLMs generally require a video-specific and adaptive solution that takes into account both video statistics and performance fluctuations across different frame conditions.

In this work, we introduce Vid-SME (Video Sharma–Mittal Entropy), the first membership inference attack specifically devised to identify videos used in the training of VULLMs. Vid-SME leverages the flexible entropy formulation of Sharma–Mittal Entropy [42, 2, 14, 51] to adaptively capture the specific inter-frame variations of video frame sequences and compute customized entropy values. To account for different frame conditions, Vid-SME further exploits the model's behavioral differences between natural and reversed frame sequences to compute the final membership score. This design is motivated by our observation that, if a video was seen during training, the model tends to predict the next token with higher confidence when frames are presented in their natural order, leading to a lower entropy value. In contrast, when processing reversed frame sequences, the model exhibits more pronounced confidence degradation on seen videos, resulting in a more noticeable increase in entropy value. This ultimately yields a larger entropy gap between natural and reversed sequences for those seen videos, which serves as a strong membership signal.

We evaluate the performance of Vid-SME on various frame conditions, target datasets and target models. The results consistently demonstrate its strong effectiveness in inferring video membership in VULLMs. We summarize our contributions as follows:

- We introduce Vid-SME, the first dedicated method for video membership inference, which adaptively adjusts the controllable parameters in Sharma–Mittal entropy and leverages reversed frame sequences to capture the inherent temporal nature and complex inter-frame variations in videos, thus achieving reliable membership inference.

- Open-sourced VULLMs are commonly trained on multi-source datasets with only a portion of the training data publicly available, making it difficult to isolate the effects of task types and data distributions on MIA performance. To enable more controlled evaluation, we establish a benchmark by training three VULLMs, each on a distinct dataset, using two representative training strategies (Video-XL [45] and LongVA [63]).

- Extensive experiments across five VULLMs (three self-trained and two open-sourced) clearly demonstrate the superiority of Vid-SME. For example, when applied to the open-sourced LLaVA-NeXT-Video-34B [27, 65], Vid-SME delivers a 28.3% improvement in AUC, an 18.1% increase in accuracy, and an impressive 293% boost in TPR@5% FPR.

## 2 Related Work

### 2.1 MultiModal Large Language Models

Building on the success of large language models (LLMs) [12, 13, 36, 52], multimodal large language models (MLLMs) [1, 11, 26, 55] integrate visual encoders to extract visual features, which are then aligned to the same dimensional space as LLM tokens through dedicated connectors, enabling effective visual-language processing. Recent advancements in MLLMs have led to significant improvements in image-related tasks. Video Understanding Large Language Models (VULLMs) [63, 28, 45, 30] further expand the capabilities of MLLMs to video understanding by encoding multi-frame features and concatenating them for uniform interpretation. The typical working pipeline of VULLMs for video data closely follows that of image-based MLLMs [30, 28]. For example, a visual encoder is usually employed to extract spatiotemporal features from videos. These features are then projected into the input space of the large language model through a learnable linear projection layer, enabling seamless integration with language tokens.

VULLMs commonly adapt pretrained image-based MLLMs for video tasks [33, 45, 63, 32, 37], which are usually trained on image and text modalities and then instruction-tuned on carefully designed video instruction data, during which only the linear projection layer is updated, while the

rest of the architecture remains frozen [32]. Recent efforts, including LongVA [63], Video-XL [45], and the LLaVA-NeXT-Video series [27, 65], focus on enhancing temporal modeling to support long video comprehension, and have demonstrated strong performance on related tasks.

## 2.2 Membership Inference Attack

Membership Inference Attacks (MIAs) [44, 4, 23, 54] aim to determine whether a specific data sample was included in a model's training set. For a machine learning model, ensuring the confidentiality of its training data is critical, as it may contain sensitive or personal information about individuals. Existing MIA methods can be broadly categorized into two types [4, 24]: metric-based and shadow model-based. Metric-based MIAs [59, 40, 49, 24] rely on evaluating certain metrics derived from the target model's outputs and making membership decisions based on predefined thresholds. In contrast, shadow model-based MIAs [44, 60] train additional models to replicate the behavior of the target model, which requires extensive computational resources and is often impractical for LLMs [24]. Thus, this work focuses exclusively on metric-based methods.

MIAs were initially applied in the context of classification models [44], but have since been extended to other types of models, such as generative models [10, 18] and embedding models [31, 47]. With the rapid advancement of LLMs and MLLMs, researchers have begun to explore the feasibility of conducting MIAs against these models as well. For example, [43] proposed Min-$K\%$, which selects the smallest $K\%$ of probabilities corresponding to the ground-truth token, while [24] argued that detecting individual images or texts is more practical in real-world scenarios and presents additional challenges. To address this, they introduced MaxRényi-$K\%$ and its variant version ModRényi, investigating the potential for extracting and attacking unimodal information from MLLMs. However, to the best of our knowledge, no existing work has explored the privacy risks of MIAs on video understanding large language models (VULLMs).

In addition, existing MIA studies on (M)LLMs can generally be categorized into those targeting pretraining data [43, 62, 8] and those targeting instruction tuning data [24, 57, 20]. Unlike these models, as dicussed in Section 2.1, VULLMs are commonly built by adapting image-based MLLMs to video tasks [33, 45, 63, 32, 37]. These models are initially trained on image–text data and instruction-tuned using video instruction datasets. As a result, MIAs against videos in VULLMs are primarily constrained to the instruction tuning stage. This highlights the unique importance of this problem: The capability of VULLMs to effectively interact with humans fundamentally depends on the instruction tuning stage, as the strength of this capability is directly tied to the quality of the instruction tuning dataset. Furthermore, developers often construct their own task-specific datasets for this stage [45, 63, 22], which introduces additional privacy risks. Motivated by these factors, our work focuses on video membership inference during the instruction tuning stage of VULLMs.

## 3 Problem Setting and Challenges

**Notation.** The token set is denoted by $\mathcal{V}$. A sequence of $L$ tokens is denoted as $X := (x_1, x_2, \ldots, x_L)$, where $x_k \in \mathcal{V}$ for $k \in [L]$. Let $X_1 \| X_2$ denote the aggregation of sequences $X_1$ and $X_2$. A video token sequence is denoted as $F_{1:T}$, where $T$ represents the number of frames. In this work, we focus on a VULLM $f_\theta$, parameterized by $\theta$, where the input of the model consists of $F_{1:T}$ and an instruction context $X_{\text{ins}}$, and the output is the response text $X_{\text{res}}$. We use $\mathcal{D}_{\text{vid}}$ to represent the video set containing the videos used in model training.

**Adversary's Goal.** We follow the standard definition of MIAs as described in [44]. Given a VULLM $f_\theta$, the adversary aims to determine whether a specific video was used during the instruction tuning stage of $f_\theta$. We formulate this attack as a binary classification problem. Let $\mathbf{A}(F_{1:T}; \theta) :\rightarrow \{0, 1\}$ denote the membership detector. During the attack, we feed the model with $F_{1:T}$ and the instruction context $X_{\text{ins}}$. The membership detector makes its decision by comparing a metric $I(F_{1:T} \oplus X_{\text{ins}}; \theta)$ with a certain threshold $\lambda$:

$$\mathbf{A}(F_{1:T}; \theta) = \begin{cases} 1 & (F_{1:T} \in \mathcal{D}_{\text{vid}}), \ \text{if} \ \ I(F_{1:T} \| X_{\text{ins}}; \theta) < \lambda, \\ 0 & (F_{1:T} \notin \mathcal{D}_{\text{vid}}), \ \text{if} \ \ I(F_{1:T} \| X_{\text{ins}}; \theta) \geq \lambda. \end{cases} \tag{1}$$

**Adversary's Knowledge.** Following the standard MIA setup [24, 57], we assume a grey-box scenario where the adversary can query the target model using the video frames and the instruction context,

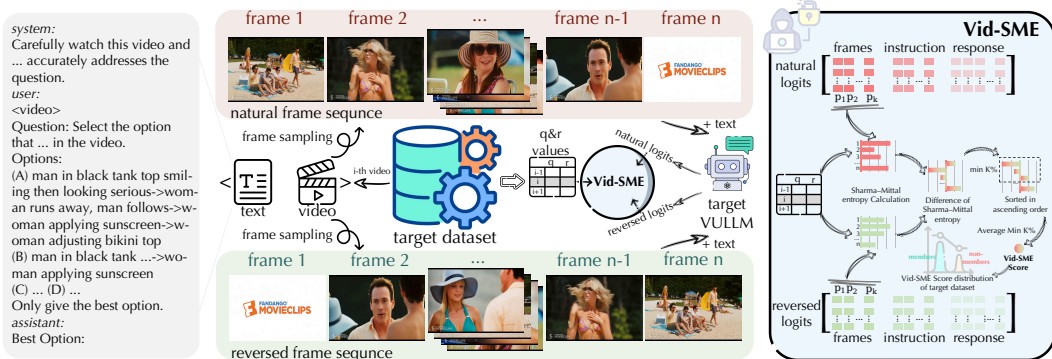

Figure 1: Vid-SME against VULLMs. **Left:** An example of the video instruction context used in our experiments. **Middle:** The overall pipeline of Vid-SME. **Right:** The detailed illustration of the membership score calculaiton of Vid-SME.

and is allowed to access the tokenizer, output logits, and generated text. However, the adversary has no knowledge of the training algorithm or the model parameters of the target model.

**Challenges. (i).** Unlike conventional LLMs and image-based MLLMs, VULLMs incorporate video modality during instruction tuning, enabling multimodal understanding beyond static images and text. The temporal nature and complex inter-frame variations inherent in video data makes membership inference significantly more challenging. **(ii).** Membership inference fundamentally relies on the model's memorization of training data [44]. However, memorization in LLMs is generally weak. This becomes more subtle for video data, where the number of frames fed into VULLMs can influence the model performance and thus influence the degree of memorization [34, 56]. The variation in the frame conditions makes the relationship between the attack performance and the model's memorization highly intricate, thereby posing additional challenges for effective attacks. **(iii).** On the dataset side, video instruction tuning data for VULLMs typically comes from diverse and heterogeneous sources [63, 45, 65], leading to highly complex data distributions. Moreover, it is often difficult to find non-member data that shares a similar distribution with the training set. Previous MIAs on text and image data attempt to synthesize non-member samples using LLMs or image generation models [24], whereas such synthesis remains challenging for video data. The distribution shift between members and non-members poses additional challenges for the evaluation of membership inference.

## 4 Method

Similar to image-based MLLMs, VULLMs also usually project the vision encoder's embedding of the video frame sequence into the feature space of LLM. Under the grey-box setting, intermediate information from the LLM is inaccessible, and gradient-based operations (e.g., backpropagation) cannot be performed. To this end, we propose a token-level video MIA that computes metrics based on the output logits at each token position.

Figure 1 illustrates the full pipeline of our proposed attack, which can be divided into three main stages: data preprocessing, model inference, and membership inference. In the data preprocessing stage, we perform frame sampling on all videos in the target dataset containing members and non-members. Without loss of generality, we adopt uniform sampling based on frame indices. Additionally, to capture the specific inter-frame variations of each video frame sequence, we customize the order parameter $q$ and deformation parameter $r$ of Sharma–Mittal Entropy [42] by incorporating the video's motion complexity and illumination variation into their determination. In the model inference stage, the sampled video frames and the instruction context are fed into the target VULLM. This step is conducted twice, using both the natural frame order and the reversed frame order, respectively. Finally, in the membership inference stage, we extract the slices of natural and reversed logits corresponding to the video frames, which can be easily located based on the model's special tokens [24]. Using the customized $q$ and $r$ values, we compute the Sharma–Mittal entropy for both slices, and derive the final membership score through the differences between the two entropy values.

**Sharma–Mittal entropy.** Sharma-Mittal entropy is one of the entropy metrics that is widely used in information theory and statistical learning due to its flexibility [42, 14, 51]. It allows for tunable sensitivity to different regions of a probability distribution, which is particularly beneficial in scenarios

involving complex distributions such as those observed in video-language modeling and membership inference attacks. It generalizes several well-known entropy formulations and can be defined as $S_{q,r}(p) = \frac{1}{1-r}\left(\left(\sum_j p_j^q\right)^{\frac{1-r}{1-q}} - 1\right), q, r \in (0, \infty) \setminus \{1\}$, where $p = \{p_j\}$ represents a probability distribution, and $q$ and $r$ are two adjustable parameters that control the entropy's sensitivity and aggregation behavior, respectively. The parameter $q$ determines how the entropy responds to the skewness of the distribution. Specifically, smaller values of $q$ increase sensitivity to low-probability (rare) events, while larger values of $q$ emphasizes more on high-probability (dominant) modes. In contrast, $r$ governs the nonlinearity and aggregation scheme in entropy calculation. Larger values of $r$ make the entropy calculation more nonlinear, thereby increasing its sensitivity to distribution peaks.

## 4.1 Adaptive Parameterization

Videos naturally exhibit highly diverse visual properties. Such diversity impacts the model's prediction distributions. For instance, fast-moving videos with large inter-frame variations often induce higher uncertainty, leading to more dispersed predictions, while stable videos result in more confident and concentrated outputs [35, 16, 67, 3]. Moreover, videos with obvious illumination variations may introduce abnormal prediction fluctuations, as sudden brightness changes can create misleading visual cues that confuse the model and cause it to overly favor certain tokens [9, 58, 46, 29]. Thus, algin with the properties of Sharma–Mittal entropy, we adapt the parameters $q$ and $r$ for each video frame sequence based on its motion complexity and illumination variation.

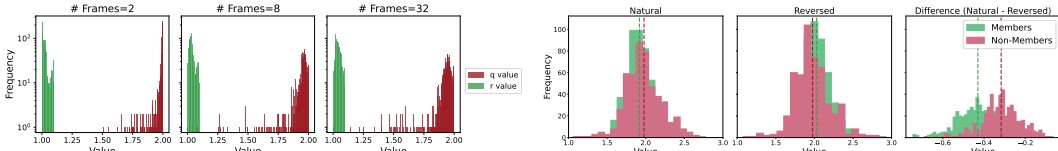

(a) Variation of $q/r$ values with respect to # frames.  (b) Natural-reversed entropy difference is essential.

Figure 2: Example of the $q/r$ value distribution and entropy distribution on Video-XL-CinePile-7B.

To do so, for the $i$-th video, after the frame sequence sampling, we quantify its motion complexity $\phi_i$ as the mean variance of optical flow [15] between consecutive frames, while its illumination variation $\lambda_i$ is measured as the standard deviation of average brightness across frames. Specifically, each frame is first converted to grayscale, and the mean brightness of each frame is computed; the standard deviation of these mean values then reflects the overall illumination variation within the sequence. Both statistics are further normalized in the target dataset to obtain normalized statistics $\hat{\phi}_i$ and $\hat{\lambda}_i$, respectively. The entropy parameters $q_i$ and $r_i$ of the $i$-th video are then determined as:

$$q_i = 1 + \beta_1 \cdot \frac{\max_j \hat{\phi}_j - \hat{\phi}_i}{\max_j \hat{\phi}_j - \min_j \hat{\phi}_j}, \quad r_i = 1 + \beta_2 \cdot \frac{\hat{\lambda}_i - \min_j \hat{\lambda}_j}{\max_j \hat{\lambda}_j - \min_j \hat{\lambda}_j}, \tag{2}$$

where $\beta_1$ and $\beta_2$ are scaling coefficients controlling the adjustment range of each parameter, which are set to 1.0 and 0.1, respectively, to better align with the nature of Sharma-Mittal entropy calculation. In this design, video frame sequences with higher motion complexity (i.e., larger $\phi$) are assigned smaller $q$ values, as higher motion typically leads to more uncertain predictions, resulting in flatter probability distributions with more low-probability tokens. Smaller $q$ values increase the sensitivity of the entropy calculation to these low-probability tokens, thus better reflecting the model's uncertainty in such cases. Meanwhile, videos with larger illumination variations (i.e., larger $\lambda$) are assigned larger $r$ values, which enhances the nonlinearity of the entropy calculation and increases its sensitivity to abnormal predictions.

## 4.2 Vid-SME

We now propose our Vid-SME, utilizing the Sharma–Mittal entropy of the next-token probability distribution. Specifically, given a token sequence $X := (x_1, x_2, \ldots, x_L)$ consisting of video frame tokens and instruction context tokens (i.e., $X = F_{1:T} \| X_{\text{ins}}$), let $p_k(\cdot) = \mathcal{P}(\cdot | x_1, \ldots, x_k; \theta)$ be the next-token probability distribution at the $k$-th position. We then extract the video-related probability slices, denoted by $\bar{p}_{1:T} = \{p_k(\cdot) \mid x_k \in F_{1:T}\}$. Accordingly, the video-related probability slices

corresponding to the reversed video sequence can be extracted, denoted as $\bar{p}_{T:1} = \{\hat{p}_k(\cdot) \mid x_k \in F_{T:1}\}$, where $F_{T:1}$ denotes the reversed video token sequence, $\hat{p}_k(\cdot)$ is the next-token probability distribution for the reversed video sequence at the $k$-th position. The Sharma-Mittal entropy is then computed for the natural and reversed probability slices, resulting in

$$S_{\text{nat}} = \{S_{q,r}\left(p_k(\cdot)\right) \mid p_k(\cdot) \in \bar{p}_{1:T}\}, \quad S_{\text{rev}} = \{S_{q,r}\left(\hat{p}_k(\cdot)\right) \mid \hat{p}_k(\cdot) \in \bar{p}_{T:1}\}, \quad (3)$$

where $S_{q,r}(\cdot)$ denotes the Sharma-Mittal entropy with adaptively determined parameters $q$ and $r$. After that, we calculate the element-wise differences between the two sequences as $\Delta S^{(\xi)} = S_{\text{nat}}^{(\xi)} - S_{\text{rev}}^{(\xi)}$, for $\xi = 1, 2, \ldots, |S_{\text{nat}}|$. Let Min-$K\%(\Delta S)$ be the smallest $K\%$ from the sequence $\Delta S$. The final Vid-SME score for current video frame sequence is computed as

$$\texttt{Vid-SME-K\%}(F_{1:T}) = \frac{1}{|\text{Min-}K\%(\Delta S)|} \sum_{\xi \in \text{Min-}K\%(\Delta S)} \Delta S^{(\xi)}. \quad (4)$$

When $K = 0$, the $\texttt{Vid-SME-K\%}$ score is defined to be $\min_\xi \Delta S^{(\xi)}$. When $K = 100$, the $\texttt{Vid-SME-K\%}$ score is the mean $\Delta S^{(\xi)}$ value of the sequence $F_{1:T}$. As $q \to 1$, the formulation of Sharma-Mittal entropy reduces to the classical Shannon entropy [41]; when $r \to 1$, Sharma-Mittal entropy reduces to Rényi entropy [39]; when $r = q$, it corresponds to Tsallis entropy [53]. Formal definitions of these simpler form can be found in Appendix F. In practice, we set a sufficiently small threshold of $1 \times 10^{-10}$. When the different between $q$ and 1, $r$ and 1, or between $q$ and $r$ falls below this threshold, the entropy calculation in Vid-SME degenerates into the corresponding simpler form.

**The variation of $q/r$ values with respect to the number of sampled frames.** As shown in Figure 2a, increasing the number of frames makes the $q/r$ values more video-specific, suggesting that the richer video information brought by additional frames is effectively reflected in the $q/r$ values.

**The significance of the natural-reversed entropy difference.** As shown in Figure 2b, while the natural and reversed entropy distributions offer some separation between members and non-members, they are insufficient for clear discrimination. In contrast, the natural-reversed entropy difference significantly amplifies the distribution gap, making the distinction much more pronounced.

## 5 Experiments

**Datasets and Models.** To comprehensively evaluate the attack performance, we construct member and non-member sets for five target models, including three self-trained models, covering various task types, video lengths, and dataset scales. The details of these datasets are summarized in Table 1. The configurations and training trajectories of the three self-trained models are given in Appendix A.

Specifically, we follow the training pipeline and model components of Video-XL-7B [45] and instruction tune the model on two distinct datasets to obtain Video-XL-NExT-QA-7B and Video-XL-CinePile-7B, respectively. NExT-QA [25] is a video question answering dataset while CinePile [38] is a video order reasoning dataset. For the NExT-QA dataset, we randomly sample 1070, 2140, and 4280 instances from both the training and testing splits to construct the member and non-member sets, respectively. This results in target datasets with three different scales (i.e., 2140, 4280 and 8560). Unless otherwise specified, we default to using 2140 instances for both members and non-members (4280 in total) in all the experiments. Results under different dataset scales are reported in Table 4. For Video-XL-CinePile-7B, the non-member set consists of all 502 instances from nine scenarios in the MLVU benchmark [66], which involve sequential reasoning tasks similar to CinePile. To ensure consistency in scale, we randomly sample 502 videos from CinePile

| Target Model | Member Data | | | | Non-Member Data | | | |
|---|---|---|---|---|---|---|---|---|
| | source | scale | duration(s) | fps | source | scale | duration(s) | fps |
| Video-XL-NExT QA-7B | NExT-QA [25] | 1070 2140 4280 | 45.22 46.01 44.68 | 28.89 28.84 28.77 | NExT-QA [25] | 1070 2140 4280 | 39.17 36.52 38.91 | 28.75 28.73 28.68 |
| Video-XL-Cine Pile-7B | CinePile [38] | 502 | 159.80 | 23.98 | MLVU [66] | 502 | 933.66 | 29.05 |
| LongVA-Capt ion-7B | Video-XL [45] | 1027 | 24.18 | 28.29 | VDC [7] | 1027 | 30.07 | 28.57 |
| LLaVA-NeXT -Video-7B/34B | Video-Instruct -100K [32] | 869 | 116.57 | 25.03 | Video-XL [45] | 869 | 24.79 | 28.02 |

Table 1: Statistics of datasets for each target VULLM.

as the member set. In addition to these, we follow the training pipeline and model components of LongVA [63] and instruction tune the model with video captioning data from Video-XL training set [45] to obtain LongVA-Caption-7B. We use all the 1027 samples from the detailed captioning category in the VDC benchmark [7] as the non-member set and randomly sample 1027 instances from the model's training set as members.

Beyond our self-trained models, we also include two open-sourced models for evaluation: LLaVA-NeXT-Video-7B [65] and LLaVA-NeXT-Video-34B [65]. For these models, we use the video

| Metric | | Video-XL-NExT-QA-7B [45] | | | Video-XL-CinePile-7B [45] | | | LongVA-Caption-7B [63] | | | LLaVA-NeXT-Video-7B [65, 27] | | | LLaVA-NeXT-Video-34B [65, 27] | | |
|---|---|---|---|---|---|---|---|---|---|---|---|---|---|---|---|---|
| | | AUC | Acc. | TPR@5% FPR | AUC | Acc. | TPR@5% FPR | AUC | Acc. | TPR@5% FPR | AUC | Acc. | TPR@5% FPR | AUC | Acc. | TPR@5% FPR |
| Perplexity | | 0.461 | 0.502 | 0.048 | 0.679 | 0.667 | 0.004 | 0.497 | 0.527 | 0.028 | 0.388 | 0.501 | 0.002 | 0.379 | 0.505 | 0.007 |
| Max_Prob_Gap | | 0.497 | 0.521 | 0.050 | 0.543 | 0.559 | 0.020 | 0.507 | 0.521 | 0.049 | 0.480 | 0.514 | 0.020 | 0.286 | 0.502 | 0.014 |
| Modified_Entropy | | 0.460 | 0.503 | 0.048 | 0.677 | 0.664 | 0.004 | 0.500 | 0.529 | 0.028 | 0.384 | 0.501 | 0.002 | 0.376 | 0.504 | 0.006 |
| Min_0% Prob | | 0.482 | 0.515 | 0.028 | 0.500 | 0.544 | 0.006 | 0.474 | 0.501 | 0.034 | 0.271 | 0.501 | 0.000 | 0.417 | 0.500 | 0.016 |
| Min_5% Prob | | 0.472 | 0.507 | 0.052 | 0.564 | 0.582 | 0.004 | 0.496 | 0.526 | 0.024 | 0.317 | 0.502 | 0.002 | 0.462 | 0.508 | 0.016 |
| Min_30% Prob | | 0.469 | 0.508 | 0.062 | 0.619 | 0.622 | 0.004 | 0.510 | 0.529 | 0.033 | 0.344 | 0.501 | 0.003 | 0.440 | 0.508 | 0.014 |
| Min_60% Prob | | 0.460 | 0.504 | 0.038 | 0.654 | 0.641 | 0.004 | 0.504 | 0.527 | 0.027 | 0.364 | 0.501 | 0.002 | 0.407 | 0.505 | 0.008 |
| Min_90% Prob | | 0.461 | 0.502 | 0.048 | 0.680 | 0.666 | 0.004 | 0.497 | 0.528 | 0.028 | 0.383 | 0.501 | 0.002 | 0.382 | 0.505 | 0.007 |
| **ModRényi** | $\alpha = 0.5$ | 0.460 | 0.504 | 0.046 | 0.694 | 0.685 | 0.006 | 0.497 | 0.526 | 0.025 | 0.398 | 0.501 | 0.002 | 0.358 | 0.503 | 0.007 |
| | $\alpha = 2$ | 0.460 | 0.501 | 0.040 | 0.703 | 0.688 | 0.006 | 0.494 | 0.524 | 0.030 | 0.411 | 0.501 | 0.002 | 0.341 | 0.504 | 0.007 |
| **Rényi** ($\alpha = 0.5$) | Max_0% | 0.457 | 0.502 | 0.036 | 0.589 | 0.616 | 0.010 | 0.521 | 0.528 | 0.032 | 0.262 | 0.500 | 0.007 | 0.378 | 0.500 | 0.010 |
| | Max_5% | 0.466 | 0.508 | 0.034 | 0.567 | 0.611 | 0.000 | 0.540 | 0.546 | 0.037 | 0.297 | 0.501 | 0.009 | 0.458 | 0.511 | 0.013 |
| | Max_30% | 0.463 | 0.511 | 0.050 | 0.575 | 0.602 | 0.002 | 0.538 | 0.547 | 0.046 | 0.251 | 0.501 | 0.006 | 0.555 | 0.557 | 0.052 |
| | Max_60% | 0.462 | 0.512 | 0.040 | 0.583 | 0.602 | 0.006 | 0.535 | 0.542 | 0.036 | 0.233 | 0.500 | 0.002 | 0.590 | 0.586 | 0.045 |
| | Max_90% | 0.469 | 0.507 | 0.052 | 0.607 | 0.617 | 0.006 | 0.496 | 0.503 | 0.044 | 0.250 | 0.500 | 0.000 | 0.585 | 0.588 | 0.044 |
| **Rényi** ($\alpha = 1$) | Max_0% | 0.450 | 0.501 | 0.024 | 0.538 | 0.579 | 0.004 | 0.506 | 0.522 | 0.028 | 0.360 | 0.500 | 0.007 | 0.383 | 0.500 | 0.012 |
| | Max_5% | 0.468 | 0.510 | 0.034 | 0.511 | 0.573 | 0.000 | 0.508 | 0.533 | 0.032 | 0.437 | 0.501 | 0.014 | 0.468 | 0.514 | 0.012 |
| | Max_30% | 0.468 | 0.509 | 0.056 | 0.563 | 0.594 | 0.000 | 0.530 | 0.538 | 0.038 | 0.312 | 0.500 | 0.001 | 0.518 | 0.546 | 0.025 |
| | Max_60% | 0.461 | 0.506 | 0.048 | 0.593 | 0.608 | 0.004 | 0.533 | 0.543 | 0.036 | 0.290 | 0.500 | 0.001 | 0.502 | 0.543 | 0.024 |
| | Max_90% | 0.460 | 0.506 | 0.044 | 0.623 | 0.620 | 0.000 | 0.514 | 0.533 | 0.037 | 0.319 | 0.500 | 0.001 | 0.479 | 0.533 | 0.021 |
| **Rényi** ($\alpha = 2$) | Max_0% | 0.470 | 0.505 | 0.012 | 0.479 | 0.553 | 0.002 | 0.484 | 0.502 | 0.035 | 0.315 | 0.500 | 0.000 | 0.403 | 0.500 | 0.010 |
| | Max_5% | 0.470 | 0.506 | 0.040 | 0.528 | 0.580 | 0.000 | 0.494 | 0.523 | 0.023 | 0.376 | 0.503 | 0.004 | 0.461 | 0.511 | 0.013 |
| | Max_30% | 0.468 | 0.510 | 0.062 | 0.591 | 0.597 | 0.002 | 0.515 | 0.532 | 0.036 | 0.341 | 0.501 | 0.001 | 0.465 | 0.520 | 0.016 |
| | Max_60% | 0.460 | 0.504 | 0.052 | 0.630 | 0.631 | 0.002 | 0.508 | 0.529 | 0.029 | 0.348 | 0.500 | 0.001 | 0.438 | 0.512 | 0.012 |
| | Max_90% | 0.460 | 0.503 | 0.052 | 0.661 | 0.655 | 0.004 | 0.499 | 0.531 | 0.028 | 0.371 | 0.500 | 0.002 | 0.416 | 0.508 | 0.010 |
| **Rényi** ($\alpha = \infty$) | Max_0% | 0.482 | 0.515 | 0.028 | 0.500 | 0.544 | 0.006 | 0.474 | 0.501 | 0.034 | 0.271 | 0.501 | 0.000 | 0.417 | 0.500 | 0.016 |
| | Max_5% | 0.472 | 0.507 | 0.052 | 0.564 | 0.582 | 0.004 | 0.496 | 0.526 | 0.024 | 0.317 | 0.502 | 0.002 | 0.462 | 0.508 | 0.016 |
| | Max_30% | 0.469 | 0.508 | 0.062 | 0.619 | 0.622 | 0.004 | 0.510 | 0.529 | 0.033 | 0.344 | 0.501 | 0.004 | 0.440 | 0.508 | 0.014 |
| | Max_60% | 0.460 | 0.504 | 0.038 | 0.654 | 0.641 | 0.004 | 0.504 | 0.527 | 0.027 | 0.364 | 0.501 | 0.002 | 0.407 | 0.505 | 0.008 |
| | Max_90% | 0.461 | 0.502 | 0.048 | 0.680 | 0.666 | 0.004 | 0.497 | 0.528 | 0.028 | 0.383 | 0.501 | 0.002 | 0.382 | 0.505 | 0.007 |
| **Vid-SME (Ours)** | Mean | 0.535 | 0.540 | 0.104 | 0.840 | 0.769 | 0.420 | 0.496 | 0.510 | 0.039 | 0.484 | 0.509 | 0.050 | 0.513 | 0.525 | 0.063 |
| | Min_0% | 0.519 | 0.528 | 0.030 | 0.559 | 0.568 | 0.114 | 0.544 | 0.545 | 0.056 | 0.490 | 0.509 | 0.050 | 0.757 | 0.692 | 0.204 |
| | Min_5% | 0.543 | 0.555 | 0.044 | 0.624 | 0.612 | 0.088 | 0.556 | 0.561 | 0.043 | 0.550 | 0.555 | 0.054 | 0.726 | 0.682 | 0.109 |
| | Min_30% | 0.545 | 0.547 | 0.074 | 0.699 | 0.670 | 0.161 | 0.541 | 0.553 | 0.050 | 0.601 | 0.579 | 0.073 | 0.696 | 0.654 | 0.102 |
| | Min_60% | 0.548 | 0.545 | 0.084 | 0.765 | 0.709 | 0.243 | 0.544 | 0.548 | 0.053 | 0.596 | 0.572 | 0.077 | 0.673 | 0.632 | 0.117 |
| | Min_90% | 0.538 | 0.541 | 0.106 | 0.830 | 0.765 | 0.305 | 0.538 | 0.550 | 0.035 | 0.523 | 0.531 | 0.058 | 0.597 | 0.577 | 0.092 |

Table 2: Results of Vid-SME and baseline methods when # frames=16. We highlight the best, second-best, and third-best results in progressively lighter shades of red, while marking the worst, second-worst, and third-worst results in progressively lighter shades of green.

caption dataset Video-Instruct-100K [32] that serves as part of their training data as the member set. Each video in this dataset has multiple questions, from which we select the one with the longest text length, resulting in 869 samples. The non-member set consists of 869 samples randomly selected from the captioning data from Video-XL training set [45].

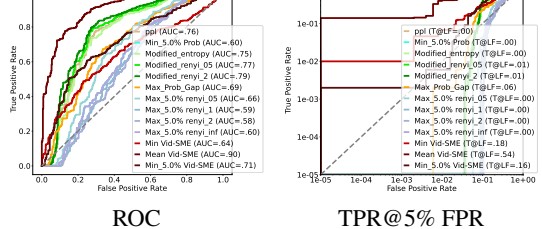

ROC      TPR@5% FPR

Figure 3: The ROC and TPR@5% FPR curves.

**Baselines.** We adopt several metric-based MIAs as baselines and compare them with Vid-SME. Specifically, we include the Loss attack [59], which corresponds to perplexity in the context of language models. We also involve the Min-$K\%$ method [43], which computes the smallest $K\%$ probabilities associated with the ground-truth tokens. We evaluate $K$ values of 0, 5, 30, 60, and 90. In addition, we adopt the Max_Prob_Gap metric [24], which captures the model's confidence by computing the difference between the maximum and the second-largest probability at each token position, followed by averaging across the sequence. We further include MaxRényi-$K\%$ and its modified variant ModRényi proposed in [24], which are specifically designed for membership inference on image-based MLLMs and utilize the Rényi entropy of next-token probability distributions. For MaxRényi-$K\%$, we set $\alpha$ to 0.5, 1, 2, and $\infty$, while for ModRényi, we use $\alpha$ values of 0.5 and 2. We also include Modified Entropy [50] as our baseline, as it is a special case of ModRényi when $\alpha \to 1$.

**Evaluation metric.** As a binary classification problem, the performance can be evaluated with the AUC score [6]. We define the members as "positive" and the non-members as "negative". We also report True Positive Rate (TPR) at low False Positive Rate (FPR) [4], which is an important metric in MIAs and measures detection rate at a meaningful threshold. We set the threshold as 5% and evaluate all the methods under TPR@5% FPR. We also report the best classification accuracy achievable by sweeping over all possible thresholds on the attack scores. Specifically, this accuracy is computed as

| # frames | Train-Test Gap |
|----------|----------------|
| 2 | 11.35 |
| 4 | 11.16 |
| 8 | 10.95 |
| 16 | 8.57 |
| 32 | 10.20 |

(a) Performance gap v.s. # frames.

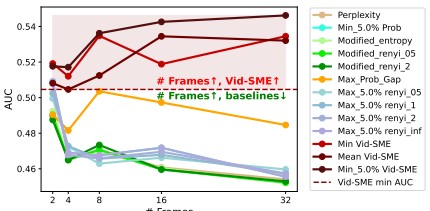

(b) Attack performance (AUC) of different methods v.s. # frames.

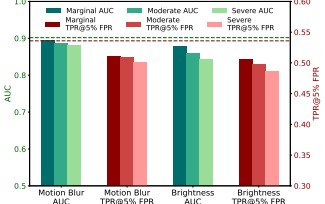

(c) AUC and TPR@5% FPR under different corruptions.

Figure 4: Analysis on: (a) Train-Test Gap v.s. # frames, (b) Attack performance v.s. # frames, (c) Attack performance under different corruption types and levels.

the maximum value of $1 - \frac{\text{FPR}+(1-\text{TPR})}{2}$ across the ROC curve, representing the optimal classification accuracy between members and non-members. We use # frames to denote the number of frames.

## 5.1 Main Results

The experimental results across the five target VULLMs are summarized in Table 2. We highlight the best, second-best, and third-best results in progressively lighter shades of red, while marking the worst, second-worst, and third-worst results in progressively lighter shades of green.

| Method | Ins. Type | Metric | | |
|--------|-----------|--------|--------|--------|
| | | AUC | Acc. | TPR@5% FPR |
| Perplexity | $I_1$ | 0.377 | 0.504 | 0.009 |
| | $I_2$ | 0.406 | 0.509 | 0.002 |
| | $I_3$ | 0.425 | 0.515 | 0.012 |
| Max_Prob_Gap | $I_1$ | 0.287 | 0.500 | 0.012 |
| | $I_2$ | 0.319 | 0.502 | 0.010 |
| | $I_3$ | 0.346 | 0.501 | 0.009 |
| Modified_Entropy | $I_1$ | 0.375 | 0.503 | 0.008 |
| | $I_2$ | 0.403 | 0.509 | 0.004 |
| | $I_3$ | 0.421 | 0.515 | 0.008 |
| Min_5.0% Prob | $I_1$ | 0.448 | 0.506 | 0.021 |
| | $I_2$ | 0.419 | 0.508 | 0.015 |
| | $I_3$ | 0.429 | 0.508 | 0.020 |
| Modified_Rényi ($\alpha = 0.5$) | $I_1$ | 0.357 | 0.503 | 0.008 |
| | $I_2$ | 0.391 | 0.506 | 0.008 |
| | $I_3$ | 0.412 | 0.512 | 0.010 |
| Max_5.0% Rényi ($\alpha = 0.5$) | $I_1$ | 0.452 | 0.506 | 0.024 |
| | $I_2$ | 0.422 | 0.509 | 0.016 |
| | $I_3$ | 0.422 | 0.505 | 0.012 |
| Min_5.0% Vid-SME | $I_1$ | 0.692 | 0.651 | 0.104 |
| | $I_2$ | 0.714 | 0.664 | 0.100 |
| | $I_3$ | 0.729 | 0.683 | 0.109 |

Table 3: Performance comparison on different instructions.

The # frames is fixed to 16. It can be observed that Vid-SME consistently achieves the best performance under all settings, especially excelling in the most critical metric, TPR@5% FPR. When the target VULLMs are Video-XL-CinePile-7B and LLaVA-NeXT-Video-7B, all baseline methods exhibit extremely low TPR@5% FPR values (around 0.001), which are impractically low for reliable membership inference. In contrast, Vid-SME consistently maintains TPR@5% FPR above 0.05 in these challenging scenarios, demonstrating remarkable improvements. In addition, the fact that baseline methods achieve AUC scores both above and below 0.5 across different settings indicates their inconsistency in distinguishing members from non-members. This suggests that they cannot serve as a reliable and unified indicator for membership inference in video-based scenarios.

To be more intuitive, we illustrate the detailed comparisons for Video-XL-CinePile-7B with $K = 0, 5, 100$ in Figure 3, which presents both ROC and TPR@5% FPR curves. Among all methods, Vid-SME variants ($K = 0, 5, 100$) consistently achieve superior performance. Notably, Vid-SME-100% reaches the highest AUC of 0.90, substantially surpassing other baselines. Additionally, Vid-SME-0% and Vid-SME-100% achieve significantly higher TPR@5% FPR (0.18 and 0.54), while most baseline methods yield almost negligible performance (close to 0.0).

**Relationship between model memorization, frame conditions and attack performance.** To analyze the relationship among model memorization, frame conditions, and attack performance, we further investigate how the train-test performance gap and attack performance change under different frame counts (# frames). Results for Video-XL-NExT-QA-7B are given in Table 4a and Figure 4b. We can observe that, when # frames are limited, the model remains uncertain on both training and test samples, leading to poor generalization but also weak memorization. Thus, although the performance gap is large, attacks are less effective as explicit memorization has not yet emerged. As # frames

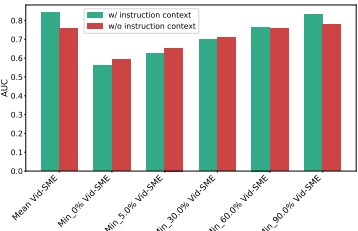

Figure 5: A comparison between with and without full context.

increase, improved understanding of video content narrows the gap, but exploitable confidence differences between members and non-members arise, enhancing the attack effectiveness. With even more frames, the model becomes highly confident on training samples while struggling with distribution shifts or increased complexity in unseen test samples, which enlarges the gap and further amplifies train-test prediction differences, making attacks highly effective. This phenomenon highlights the non-linear relationship between memorization and MIA vulnerability in the context of VULLMs.

## 5.2 Ablation Study

**Infleunce of instruction context.** We now refer to the instruction context used in our main experiments as $I_1$. To explore the influence of instruction context, we design two alternative contexts, denoted as $I_2$ and $I_3$. The details of $I_{1,2,3}$ are provided in Appendix C. The results when # frames = 8 and the target model is LLaVA-NeXT-Video-34B are reported in Table 3. As shown, the impact of the contexts is not significant. Furthermore, we observe that Vid-SME is less sensitive to context variations compared to other baselines, indicating better stability in its attack effectiveness.

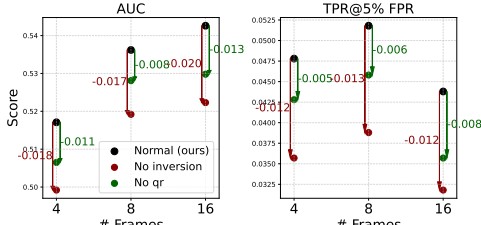

Furthermore, we investigate the scenario where text provides minimal information, and video frames are combined with only a short query text before being fed into the model, which makes the attack results independent of video-related understanding tasks. The

Figure 6: A comparison between with and without full instructions.

short query text used here can be found in Appendix C. The results under this setting, using Video-XL-CinePile-7B as the target model and # frames = 8, are reported in Figure 5. It can be observed that when the selection range of $\Delta S$ is small (e.g., $K = 0, 5, 30$), attacks using only the short query text outperform those using the full instruction context. However, as the selection range of $\Delta S$ becomes more representative of the overall video probabilities (i.e., $K \uparrow$), the attack performance with the full instruction context gradually surpasses that of the short query text. This observation aligns with intuition: when the prediction probabilities for video-related tokens are shaped by rich task-specific textual context, the model's response generation becomes more strongly grounded in the video frames, increasing its reliance on the complete visual information. Overall, however, the performance difference between the two is not substantial.

**Importance of $q/r$ adaptation and reverse frame sequence.** We further present the results when disabling the adaptive $q/r$ values (No qr) and when removing the reversed frame sequence when calculating the membership score (No inversion). For the former, we assign fixed $q$ and $r$ values (i.e., $q = 2.0$, $r = 1.0$) across the entire target dataset. For the latter, instead of using $\Delta S$, we directly adopt $S_{nat}$ to compute the final membership score. The target model in this experiment is Video-XL-NExT-QA-7B, and we report the Vid-SME-5% results. As shown in Figure 6, removing either component leads to a significant performance drop, demonstrating the

| Method | # Frames | Dataset Scale | AUC | Acc. | TPR@5% FPR |
|---|---|---|---|---|---|
| Mean Vid-SME | 8 | 2140 | 0.524 | 0.530 | 0.071 |
| | | 4280 | 0.531 | 0.541 | 0.064 |
| | | 8560 | 0.526 | 0.533 | 0.043 |
| | 16 | 2140 | 0.537 | 0.545 | 0.078 |
| | | 4280 | 0.535 | 0.540 | 0.104 |
| | | 8560 | 0.539 | 0.562 | 0.075 |
| Min_30.0% Vid-SME | 8 | 2140 | 0.522 | 0.534 | 0.050 |
| | | 4280 | 0.527 | 0.538 | 0.041 |
| | | 8560 | 0.538 | 0.539 | 0.082 |
| | 16 | 2140 | 0.548 | 0.541 | 0.064 |
| | | 4280 | 0.545 | 0.547 | 0.074 |
| | | 8560 | 0.542 | 0.545 | 0.079 |

Table 4: Performance comparison on different instructions.

critical role of $q/r$ adaptation and reversed frame sequence in performing membership inference attacks against VULLMs.

**Influence of video frame corruptions.** The motivation is to detect whether videos are used in training even under potential video corruption. In Figure 4c, we report the attack performance under two different corruptions (Motion Blur and Brightness) at three different levels of corruptions (Marginal, Moderate and Severe). Detailed corruption parameters and examples of the corrupted video frames are given in Appendix B. It can be observed that corrupted video frames make MIAs more difficult, but members can still be detected successfully.

**Influence of dataset scales.** Table 4 presents the attack performance of Vid-SME on Video-XL-NExT-QA-7B under varying dataset scales. The results show that Vid-SME remains consistently effective as the dataset size increases, demonstrating its scalability.

## 6 Conclusion

In this work, we investigate the membership inference risk in video understanding large language models (VULLMs). We propose Vid-SME, the first membership inference attack tailored for VULLMs, and self-train three VULLMs for more comprehensive evaluation. Unlike existing methods that fail to capture video-specific temporal dependencies, Vid-SME leverages an adaptive parameterization strategy and both natural and reversed frame sequence to compute the Sharma–Mittal entropy for robust membership signals. Extensive experiments demonstrate the strong effectiveness of Vid-SME.

## Acknowledgement

This project is supported by the National Research Foundation, Singapore, and Cyber Security Agency of Singapore under its National Cybersecurity R&D Programme and CyberSG R&D Cyber Research Programme Office (Award: CRPO-GC1-NTU-002).

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

## A  Model Configurations and Training States

We report the model configurations of the three self-trained VULLMs in Table 5, and the training loss and gradient norm over steps in Figure 7.

| Field | Video-XL-NExT-QA-7B | Video-XL-CinePile-7B | LongVA-Caption-7B |
|---|---|---|---|
| Base LLM | Qwen2-7B-Instruct-224K | Qwen2-7B-Instruct-224K | Qwen2-7B-Instruct-224K |
| bos_token_id | 151643 | 151643 | 151643 |
| eos_token_id | 151645 | 151645 | 151645 |
| Hidden Activation | silu | silu | silu |
| Resampler type | spatial_pool | spatial_pool | spatial_pool |
| Vision tower | clip-vit-large-patch14-336 | clip-vit-large-patch14-336 | clip-vit-large-patch14-336 |
| Vision tower lr | 2e-6 | 2e-6 | 2e-6 |
| Max window layers | 28 | 28 | 28 |
| Projector type | mlp2x_gelu | mlp2x_gelu | mlp2x_gelu |
| #Heads | 28 | 28 | 28 |
| #Hidden Layers | 28 | 28 | 28 |
| KV heads | 4 | 4 | 4 |
| Tokenizer padding side | right | right | right |
| Vocab size | 152064 | 152064 | 152064 |
| Added tokens | <lendoftextl>: 151643 <lim_endl>: 151645 <lim_startl>: 151644 pad_token: <lendoftextl> | <lendoftextl>: 151643 <lim_endl>: 151645 <lim_startl>: 151644 pad_token: <lendoftextl> | <lendoftextl>: 151643 <lim_endl>: 151645 <lim_startl>: 151644 pad_token: <lendoftextl> |
| Special tokens map | eos_token: <lim_endl> | eos_token: <lim_endl> | eos_token: <lim_endl> |

Table 5: Model configurations of the three self-trained models.

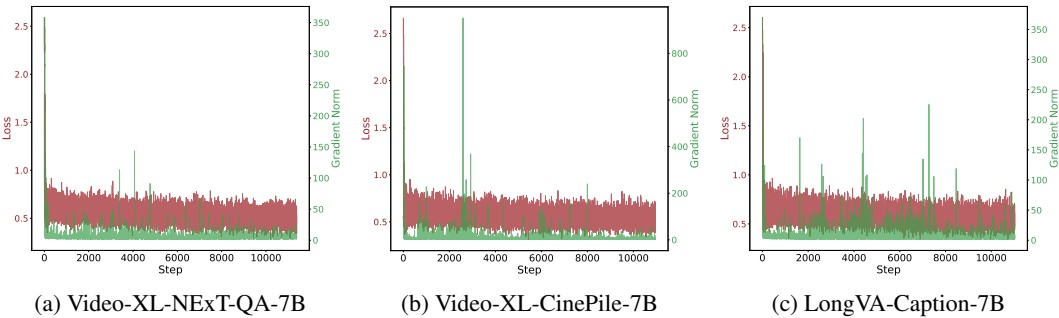

(a) Video-XL-NExT-QA-7B         (b) Video-XL-CinePile-7B         (c) LongVA-Caption-7B

Figure 7: Training Loss and Gradient Norm over Steps for the three self-trained models.

## B  Examples of the Video-Text Instruction Context

In Figure 8, we give an example of the video-text instruction context used in our experiments. In addition, we also provide the corrupted video frames under different types and levels of corruptions in Figure 8. The details of the parameters of different corruptions are given in Table 6. Specifically, for brightness corruption, we adjust the pixel intensity by randomly adding/subtracting a constant value of 20, 60, and 100 for marginal, moderate, and severe conditions, respectively. For motion blur, we apply a convolutional kernel with size and angle parameters set to (10, 5), (15, 5), and (20, 10) to simulate increasing degrees of blur under the same three corruption levels.

| | Brightness | Motion Blur |
|---|---|---|
| Marginal | 20 | (10,5) |
| Moderate | 60 | (15,5) |
| Severe | 100 | (20,10) |

Table 6: Brightness and Motion Blur Levels under Different Conditions.

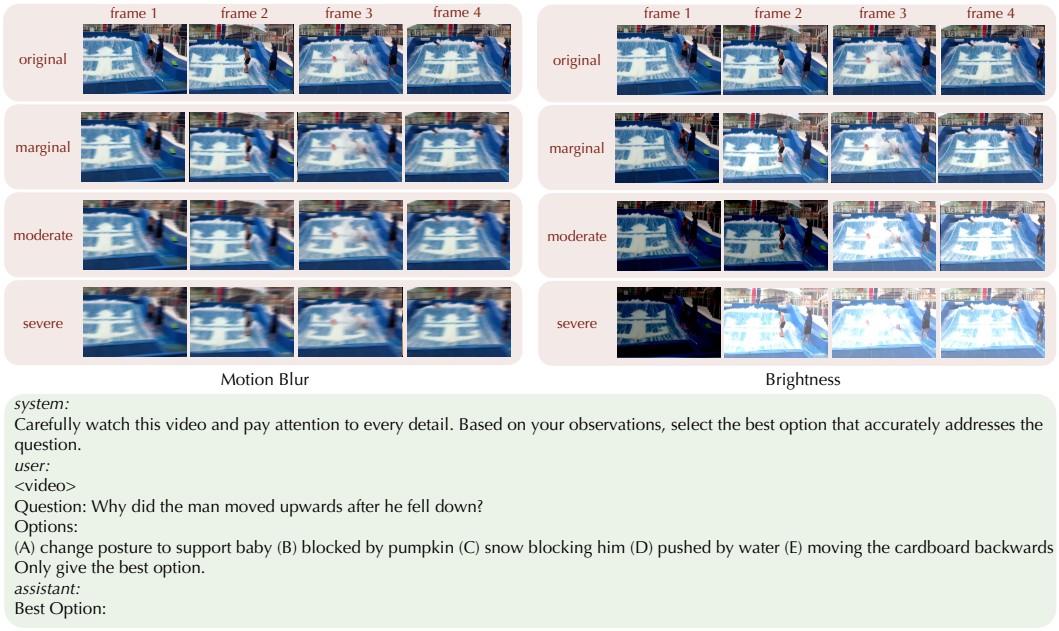

Figure 8: Example of the video-text instruction context under different types and levels of corruptions.

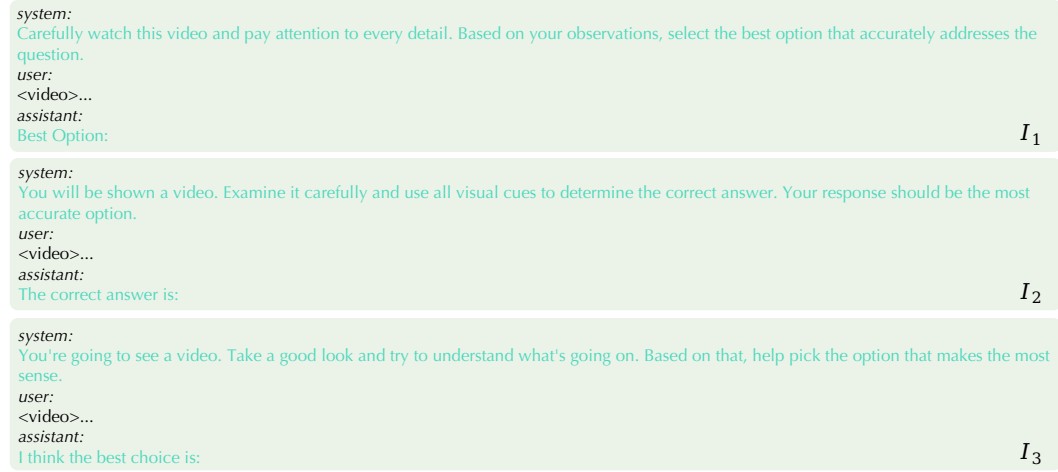

Figure 9: The three different instruction contexts used in the ablation study.

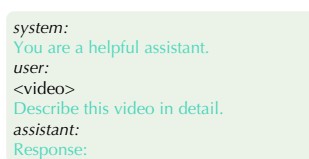

Figure 10: The short query text used in the ablation study.

## C  Different Instructions Used in the Ablation Study.

We give the contents of the three different instructions used in the ablation study in Figure 9. The short query text used in the ablation study is given in Figure 10.

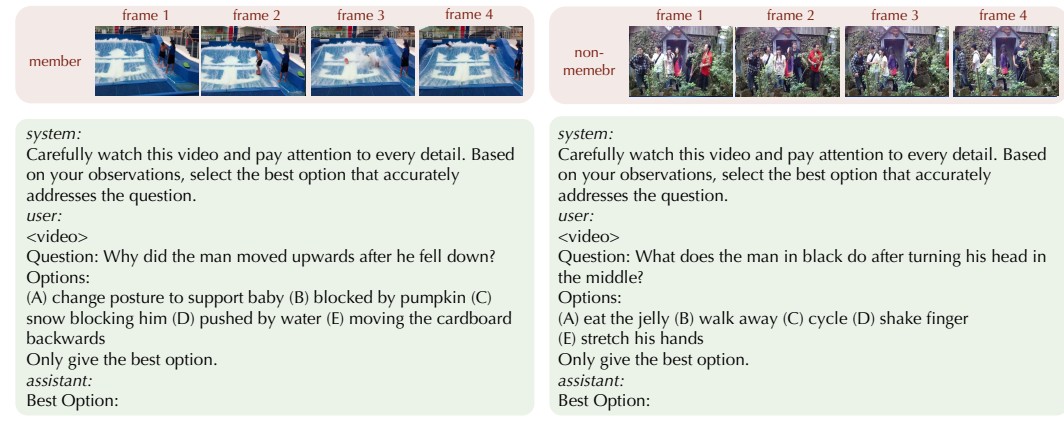

Figure 11: An example of member and non-member data for Video-XL-NExT-QA-7B.

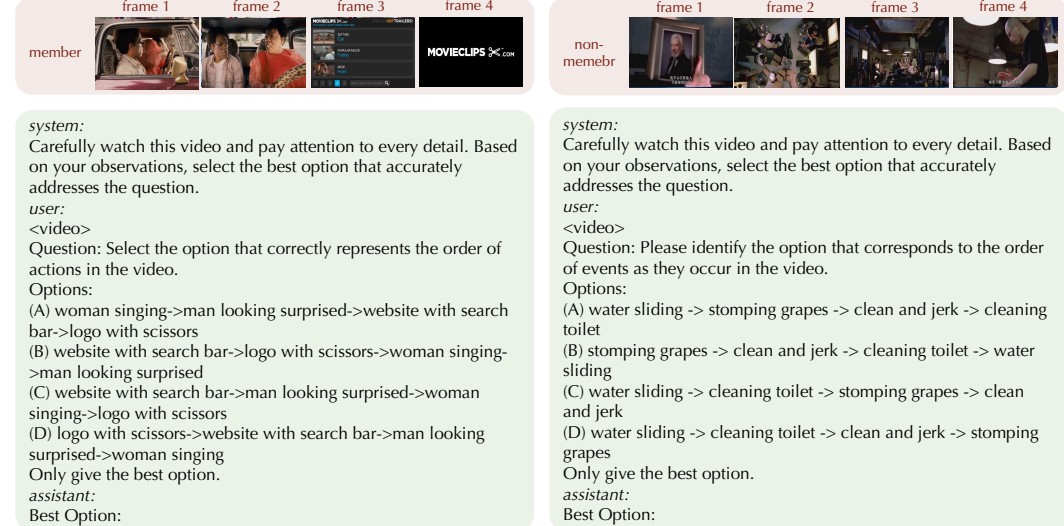

Figure 12: An example of member and non-member data for Video-XL-CinePile-7B.

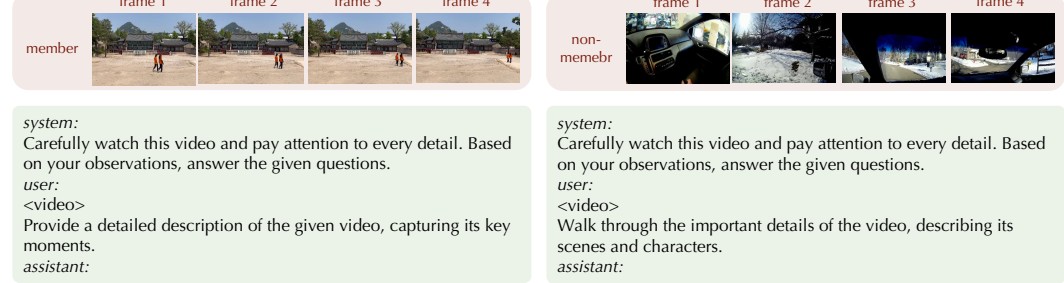

Figure 13: An example of member and non-member data for Longva-Caption-7B.

# D  Computation resource usage.

The three self-trained models are trained on 8 A100 GPUs, while all experiments are conducted using 8 NVIDIA RTX A5000 GPUs.

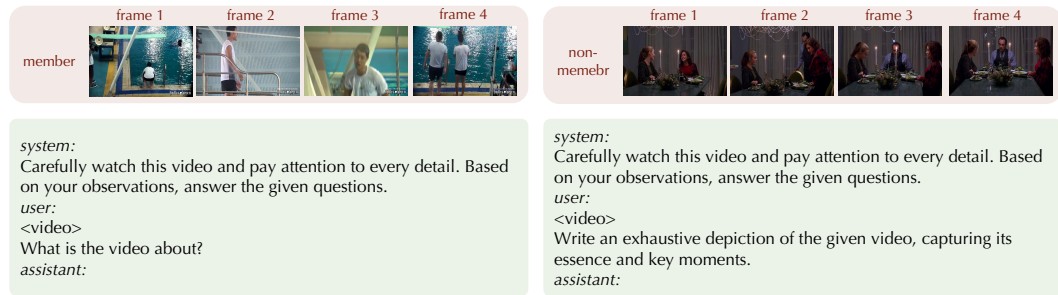

Figure 14: An example of member and non-member data for LLaVA-NeXT-Video-7B/34B.

# E Examples of Members and Non-Members of the Target Models.

We give examples of members and non-members of the five target models used in our experiments in Figures 11, 12, 13, 14.

# F Simplified Terms of Sharma–Mittal Entropy

The Sharma–Mittal entropy for a probability distribution $p = \{p_i\}$ is defined as:

$$S_{q,r}(p) = \frac{1}{1-r}\left[\left(\sum_i p_i^q\right)^{\frac{1-r}{1-q}} - 1\right], \tag{5}$$

where $q$ controls the sensitivity to distribution skewness, and $r$ determines the nonlinearity of aggregation. This generalized formulation subsumes several classical entropy measures as special cases. We give the formal definitions as follows:

## F.1 Reduction to Shannon Entropy

As both $q \to 1$ and $r \to 1$, Equation (5) reduces to Shannon entropy:

$$\lim_{q \to 1, r \to 1} S_{q,r}(p) = -\sum_i p_i \log p_i. \tag{6}$$

This limit follows from applying L'Hôpital's Rule to both the exponent and denominator as $q \to 1$ and $r \to 1$.

## F.2 Reduction to Rényi Entropy

When $r \to 1$ and $q \neq 1$, Equation (5) simplifies to the Rényi entropy:

$$\lim_{r \to 1} S_{q,r}(p) = \frac{1}{1-q} \log \sum_i p_i^q. \tag{7}$$

## F.3 Reduction to Tsallis Entropy

When $r = q$, Equation (5) reduces to the Tsallis entropy:

$$S_{q,q}(p) = \frac{1}{1-q}\left(\sum_i p_i^q - 1\right). \tag{8}$$

These reductions demonstrate that Sharma–Mittal entropy serves as a unified framework encompassing Shannon, Rényi, and Tsallis entropies as limiting cases.

# G  Broader Impact

This work explores the privacy vulnerabilities of Video Understanding Large Language Models (VULLMs) through membership inference attacks (MIAs), revealing how sensitive training data can be partially reconstructed or identified from model behavior.

## G.1  Positive Societal Impacts

Our study contributes to the broader goal of trustworthy multimodal AI by uncovering potential privacy risks before such models are widely deployed. The insights and benchmark we introduce can inform defensive research, such as differential privacy mechanisms, data sanitization, and auditing protocols, helping model developers detect and mitigate leakage risks in future VLM releases. Moreover, the framework provides a systematic methodology for assessing privacy robustness in multimodal systems, filling a critical gap between vision and language privacy research.

## G.2  Negative Social Impact

At the same time, the attack methods demonstrated in this paper could, if misused, be exploited to extract private or copyrighted visual content from proprietary VLMs. Although our experiments are conducted under controlled academic settings, similar approaches might be repurposed for malicious data mining or surveillance.

