# OpenReview forum: "Vid-SME: Membership Inference Attacks against Large Video Understanding Models"
_NeurIPS.cc/2025/Conference — NeurIPS 2025 poster_

### Official Review · Reviewer_7wvH · 2025-06-30

**Clarity:** 2
**Significance:** 3
**Originality:** 3
**Rating:** 5
**Confidence:** 2

**Summary:**

Authors explore the task inferring if a given video was used at the instruction-tuning stage of video understanding LLMs (VULLMs), referred to as membership inference attack. They use a form of entropy, Sharma–Mittal entropy, and calculate its difference between natural and reversed video sequences, with some adaptive parameters. The calculation is performed using the probability of the next video token given past, for the videos used. Thorough experiments using multiple MLLMs highlight the performance.

**Questions:**

See weaknesses.

**Ethical Concerns:**

["NO or VERY MINOR ethics concerns only"]

**Final Justification:**

Authors resolve all concerns. Raising vote to Accept.

**Limitations:**

Yes

**Quality:**

2

**Strengths And Weaknesses:**

**Strengths**
1. Clever idea of entropy calculation well suited for video tasks and well-motivated from prior video works
2. Novel exploration into inference attacks specific for video LLMs

**Weaknesses**
1. How does this work with newer generic MLLMs like Qwen2.5-VL (trained on image and video)? Comparisons are against relatively older video LLMs.
2. What is the inference time? How costly is the SM-entropy calculation? This is missing.
3. The method appears sensitive to frame count. What is the exact sampling rate of the video used during training is unknown?

---

> ### Author Rebuttal · Authors · 2025-07-28
>
> It’s our great honor to receive Reviewer 7wvH’s thoughtful comments and kind support for our work. We would like to address the questions reflected in the review below.
>
> ---
>
> > **[W1]**:  How does this work with newer generic MLLMs like Qwen2.5-VL?
>
> **A1:** Thanks for the valuable comment. We further conduct an experiment on **Qwen2.5-VL-7B**:
>
> * We follow the scripts in its official codebase and use **the generic subset of lmms-lab/VideoChatGPT** as the member data (randomly sampling 1027 examples to match the number of non-members). For the non-member set, we use all 1027 samples from **the detailed captioning category in the VDC benchmark**. The number of frames is set to 8 and 16. We report a detailed comparison of attack performance between Vid-SME and Rényi entropy. The results are presented in the tables below.
>
> _\# Frames = 8:_
>
> | Method         | Variant   | AUC   | Acc.  | TPR@5% FPR |
> |----------------|-----------|-------|-------|-------------|
> | Rényi (α=2)           | Max_0%    | 0.403 | 0.500 | 0.009  |
> |                | Max_5%    | 0.442 | 0.504 | 0.012       |
> |                | Max_30%   | 0.487 | 0.508 | 0.014       |
> |                | Max_60%   | 0.506 | 0.516 | 0.018       |
> |                | Max_90%   | 0.510 | 0.521 | 0.004       |
> | Vid-SME        | Mean      | 0.576 | 0.568 | 0.124   |
> |                | Min_0%    | 0.629 | 0.614 | 0.176       |
> |                | Min_5%    | 0.653 | 0.625 | 0.199       |
> |                | Min_30%   | **0.674** | **0.643** | 0.226       |
> |                | Min_60%   | 0.658 | 0.629 | 0.231       |
> |                | Min_90%   | 0.655 | 0.621 | **0.234** |
>
> _\# Frames = 16:_
>
> | Method         | Variant   | AUC   | Acc.  | TPR@5% FPR |
> |----------------|-----------|-------|-------|-------------|
> | Rényi (α=2)            | Max_0%    | 0.418 | 0.503 | 0.016       |
> |                | Max_5%    | 0.473 | 0.509 | 0.017      |
> |                | Max_30%   | 0.501 | 0.514 | 0.020       |
> |                | Max_60%   | 0.512 | 0.522 | 0.014       |
> |                | Max_90%   | 0.509 | 0.518 | 0.018       |
> | Vid-SME        | Mean      | 0.594 | 0.586 | 0.128 |
> |                | Min_0%    | 0.642 | 0.628 | 0.194       |
> |                | Min_5%    | 0.672 | 0.642 | 0.231       |
> |                | Min_30%   | **0.693** | **0.685** | 0.274       |
> |                | Min_60%   | 0.665 | 0.630 | **0.276**       |
> |                | Min_90%   | 0.668 | 0.627 | 0.248 |
>
> The results indicate that Vid-SME consistently achieves superior performance on recent models such as Qwen-2.5 VL, while maintaining its scalability with respect to the number of frames.
>
> ---
>
> > **[W2]**: What is the inference time? How costly is the SM-entropy calculation?
>
> **A2:** Thanks for raising this point. In the table below, we report a detailed comparison of inference time per video and attack performance between Rényi entropy and Vid-SME. The experiments are conducted using the Video-XL-CinePile-7B model with frame numbers set to 8 and 16, and with both methods using the variant of Max_90%/Min_90% and α in Rényi entropy is set to 2 (note that the variant used and the α value in Rényi entropy do not affect the inference time).
>
> _\# Frames = 8:_
>
> | Method        | Variant   | AUC   | Acc.  | TPR@5% FPR | Time per Video (s) |
> |---------------|-----------|-------|-------|-------------|---------------------|
> | Rényi (α=2) | Max_90%   | 0.559 | 0.585 | 0.004       |        6.11             |
> | Vid-SME    | Min_90%   | 0.806 | 0.760 | 0.289 |       9.45              |
>
> _\# Frames = 16:_
>
> | Method        | Variant   | AUC   | Acc.  | TPR@5% FPR | Time per Video (s) |
> |---------------|-----------|-------|-------|-------------|---------------------|
> | Rényi (α=2)  | Max_90%   | 0.661 | 0.655 | 0.004       |        11.67             |
> | Vid-SME    | Min_90%   | 0.830 | 0.765 | 0.305 |       16.76              |
>
> * Compared to Rényi entropy, the additional computational cost of Vid-SME lies in **the computation of the adaptive q/r values**. When these video-specific values are not computed, the inference-time cost of Vid-SME is nearly identical to that of Rényi entropy, as **both require two forward queries to the model**.
>
> * As the number of frames increases, Vid-SME demonstrates **good scalability in both inference time and performance improvement**.
>
> We sincerely hope that our responses adequately address the concern and we will include the discussion in our revision.
>
> ---
>
> > **[W3]**: The method appears sensitive to frame count. What is the exact sampling rate of the video used during training is unknown?
>
> **A3:** Thanks for the valuable comment. Regarding the influence of frame count, we would like to clarify that this is **not a method-related property but rather an intrinsic property of video data**—varying the number of frames naturally leads to different degrees of temporal information being captured. As shown in the results provided in our responses to [W1] and [W2], Vid-SME **exhibits consistent scalability in terms of inference time and attack performance as the number of frames varies**. For the three self-trained VULLMs, we adopted a training frame count of 128. We sincerely hope our responses help clarify this point and we will include these clarifications and relevant details in our revision.
>
> ---
>
> Thanks again for the thoughtful and constructive feedback. If reviewer 7wvH has any remaining concerns, we would definitely love to clarify further.

---

> > ### Comment · Reviewer_7wvH · 2025-08-02
> >
> > Thanks for the detailed response resolving all concerns. Raising the vote to accept.

---

> > > ### Author Response · Authors · 2025-08-03
> > > **Thanks for agreeing to raise the score**
> > >
> > > We are more than encouraged to hear that our response has resolved all the concerns of Reviewer 7wvH, and **we sincerely appreciate reviewer 7wvH’s decision to further raise the score** to 'accept', this is truly a deeper and stronger affirmation of our work. Thanks again for reviewer 7wvH’s time and insightful feedback throughout the review process.

---

### Official Review · Reviewer_JbwM · 2025-06-30

**Clarity:** 3
**Significance:** 3
**Originality:** 3
**Rating:** 5
**Confidence:** 4

**Summary:**

This paper proposes a novel MIAs method used in video understanding LLMs (VULLMs), termed Vid-SME (Video Sharma-Mittal Entropy). It leverages an adaptive parameterization strategy and both natural and reversed sequences to compute the Sharma-Mittal entropy, which effectively captures the inherent temporal variations for model behavior differences. As such, Vid-SME obtains the robust membership scores and shows strong effectiveness on various self-trained and open-sourced VULLMs.

**Questions:**

Refer to the Weaknesses. I will raise the score if well addressing these Weaknesses.

**Ethical Concerns:**

["NO or VERY MINOR ethics concerns only"]

**Final Justification:**

My concerns are well addressed in the rebuttal, and I keep my original score to accept this paper.

**Limitations:**

Yes

**Quality:**

3

**Strengths And Weaknesses:**

Strengths:
1. Well-motivated and well-written in this paper.
2. A strong extension of the proposed Sharma-Mittal Entropy to decide the final Vid-SME score, i.e., Sharma-Mittal Entropy is a general version of the previous Renyi entropy.
3. Superior performance achieved by the proposed Vid-SME on various VULLMs for MIAs, as well as a sufficient ablation study to evaluate the effectiveness of the proposed modules.

Weaknesses:
1. The idea of Natural and reversed sequences was used for Video Anomaly Detection, such as the reference [1]. Except for the task difference, what are the main benefits and differences to the work [1]. In addition, it is unclear to know whether the previous baselines (e.g., Rényi) use the information of natural and reversed sequences. If not, how to ensure the fairness of the comparison?
2. The threshold to determine whether the member data is important to performance, while this paper lacks sufficient experiments on the selection of this threshold.

[1] Comprehensive regularization in a bi-directional predictive network for video anomaly detection. In AAAI, 2022.

---

> ### Author Rebuttal · Authors · 2025-07-28
>
> It’s our great honor to receive Reviewer JbwM’s thoughtful comments and strong support for our work. We would like to address the concerns as below.
>
> ---
>
> > **[W1.1]**:  The main benefits and differences to the mentioned work except for the task difference.
>
> **A1.1:** Thanks for bringing this work to our attention. [1] is indeed a valuable contribution that proposes a self-supervised temporal discrimination framework for VAD. We would like to provide the following perspectives:
>
> * In the mentioned work [1], **the model is explicitly trained** to distinguish between forward and reversed videos in order to capture the temporal dynamics of normal behavior, thereby enabling the identification of abnormal segments. In contrast, Vid-SME **does not involve training any additional temporal classifier**. Instead, we directly utilize the model’s response to natural-reversed videos as a controlled signal to probe a deployed VULLM's internal behavioral asymmetries—specifically, how its confidence distribution shifts for member/non-member videos.
>
> * Furthermore, [1] is fundamentally **video-centric**, focusing solely on the features intrinsic to the video itself, **without involving interaction with downstream VULLMs trained on it**. In comparison, Vid-SME targets the **video–model interplay**, leveraging both **the properties of the video and the model’s reaction to it**. This distinction reflects a deeper emphasis on how video content is encoded, retained, and reflected within the model.
>
> We sincerely hope that our responses provide a reasonable comparison between the two and we will add the above discussions into the related work section of our revision.
>
> [1] Comprehensive regularization in a bi-directional predictive network for video anomaly detection. In AAAI, 2022.
>
> ---
>
> > **[W1.2]**:  Whether the previous baselines (e.g., Rényi) use the information of natural and reversed sequences. If not, how to ensure the fairness of the comparison.
>
> **A1.2:** Thanks for raising this point. We followed **the default pipeline of the baseline methods** as described in their original papers without any modifications, ensuring a fair and consistent evaluation. In addition, we would like to point out that Rényi entropy also requires querying the source model twice.
>
> Motivated by the comment, we conduct an additional experiment by adapting Rényi entropy to incorporate the natural–reversed video frames. Specifically, we use **the difference between the Rényi entropies of the natural and reversed frames as the attack signal** (termed as Rényi (w), now the calculation of Rényi entropy for each video need to query/forward the model 4 times), and evaluate whether this modification improves the inference performance. The α in Rényi entropy is set to 2 and the frame number is set to 16. The results with both methods using the variant of Max_90%/Min_90% are presented below.
>
> | Method        | Variant   | AUC   | Acc.  | TPR@5% FPR | Time per Video (s) |
> |---------------|-----------|-------|-------|-------------|---------------------|
> | Rényi (w/o)  | Max_90%   | 0.661 | 0.655 | 0.004       |        11.67             |
> | Rényi (w)  | Max_90%   | 0.717 | 0.693 | 0.127       |        24.59             |
> | Vid-SME    | Min_90%   | 0.830 | 0.765 | 0.305 |       16.76              |
>
> As shown, the modified attack achieves a performance improvement over the original setting that does not leverage the temporal information of the video. This highlights the **general effectiveness of temporal informations** in enhancing the attack signal.
>
> ---
>
> > **[W2]**:  The threshold to determine the member data.
>
> **A2:** Thanks for pointing this out. In our evaluation, we follow the standard practice in membership inference attacks by **not fixing a threshold per sample**, but instead evaluating the attack’s performance **across all possible thresholds** using metrics such as AUC. AUC reflects the trade-off between true positive and false positive rates over the entire threshold range, providing a threshold-independent and robust assessment of the method's effectiveness. We will make the description clearer in the revision.
>
> ---
>
> Thanks again for the thoughtful and constructive feedback. If reviewer JbwM has any remaining concerns, we are happy to clarify further.

---

> > ### Comment · Reviewer_JbwM · 2025-08-06
> >
> > Thank you for your response. My concerns are well addressed in this rebuttal. I keep my original rate to accept this paper.

---

> > > ### Author Response · Authors · 2025-08-06
> > > **Thanks for the support**
> > >
> > > We are more than encouraged to hear that our response has well addressed the concerns of Reviewer JbwM, and **we are truly greatful for reviewer JbwM's strong support for our work**. Thanks again for reviewer JbwM's thoughtful and constructive feedback throughout the review process.

---

### Official Review · Reviewer_CyeB · 2025-07-02

**Clarity:** 3
**Significance:** 3
**Originality:** 3
**Rating:** 5
**Confidence:** 3

**Summary:**

The paper presents Vid-SME (Video Sharma-Mittal Entropy) for membership inference attack on video understanding LLMs (VULLMs). The key idea is to use Sharma-Mittal Entropy to measure the token prediction difference between a normal input video and its reverse version, and consequently predict if the video belongs to the training set of the VULLM.

**Questions:**

See weaknesses.

**Ethical Concerns:**

["NO or VERY MINOR ethics concerns only"]

**Final Justification:**

The rebuttal addresses my main concerns, so I raised the rating to Accept.

**Limitations:**

Yes.

**Paper Formatting Concerns:**

None.

**Quality:**

3

**Strengths And Weaknesses:**

Strengths:

According to the paper, existing membership inference attack studies are limited to image LLMs, and hence the presented work is novel for video understanding LLMs.

The idea of measuring the token prediction difference between a normal video and its reverse version is interesting and intuitively makes sense.

The use of Sharma-Mittal entropy is novel to this reviewer’s knowledge.

The benchmark contribution may provide useful facilities for future research on this topic.


Weaknesses:

The problem of membership inference attack seems to be very impractical in the current setting. The graybox assumption, i.e., known token prediction but not other model information sounds quite artificial.

While the motivation of comparing a normal video and its reversed version is interesting, is this actually true for videos both inside and outside the training set? In particular, for VULLM that have seen tons of videos, it likely already learns general time-of-arrow pattern.

In Line 132, it says that “… whether a specific video was used during the instruction tuning state …”. What if the input video has been used in the pretraining stage? Will the method still work?

Line 135, the notation of I(…) is not consistent with that in equation (1).

---

> ### Author Rebuttal · Authors · 2025-07-28
>
> It's our great honor to have reviewer CyeB's valuable comments and kind support for our work. We would like to response to the questions as below.
>
> ---
>
> > **[W1]**: The practicality of membership inference attacks under gray-box scenario.
>
> **A1:** Thanks for the comment. We would like to clarify that this scenario is practically motivated and commonly adopted in previous works (e.g., Min-K%, Rényi entropy). This is attributed to its broad applicability in real-world scenarios:
>
> * Many closed-source model APIs such as OpenAI's GPT-4 API **expose top-five per-token probabilities**, which also can be seen as a gray-box scenario in our context and the adversaries can exploit this information to conduct attacks. We follow this setting (where only the top-five per-token probabilities are accessible) and conduct an experiment on Video-XL-CinePile-7B. As shown in the table below, Vid-SME remains reliable in performing membership inference under this partial-knowledge scenario.
>
> | Method        | Variant   | AUC   | Acc.  | TPR@5% FPR |
> |---------------|-----------|-------|-------|-------------|
> | Vid-SME    | Mean   | 0.775 | 0.703 | 0.305 |
> || Min_60%   | 0.657 | 0.628 | 0.229 |
>
>
> * Moreover, the gray-box setup represents a **conservative risk lower-bound**. If an attack succeeds under the gray-box setting, it indicates that the system may already be operating at the boundary of its privacy guarantees.
>
> * Furthermore, such attacks are not only relevant for external attackers, but also valuable for **model developers/owners to audit and monitor their own models for potential privacy risks**.
>
> We sincerely hope that our responses can alleviate this concern and we will further clarify our practical motivations in the revision.
>
>
> ---
>
> > **[W2]**: While the motivation of comparing a normal video and its reversed version is interesting, is this actually true for videos both inside and outside the training set?
>
> **A2:** Thanks for highlighting this important consideration. We believe the observed phenomenon reflects a generalizable statistical signal of VULLM memorization.
>
> * This is conceptually analogous to previous membership inference works in text and image domains, which also leverage statistical disparities between members and non-members even though models have seen large-scale text and image data.
>
> * In our case, we agree with the reviewer that the time-to-arrow pattern may already be well preserved by the model. Nevertheless, the model demonstrates **varying degrees of responsiveness to this pattern between seen and unseen videos**.
>
> * This discrepancy can be reliably captured via entropy calculations, as supported by our empirical findings: for videos seen during training, the model tends to **internalize more rigid temporal representations**. When these temporal patterns are violated by reversing the video frames, the model exhibits **asymmetric behavioral responses (e.g., sharper confidence drops), which are less prominent in non-member videos**.
>
> We sincerely hope our responses help clarify this point and we will include the discussion in our revision.
>
> ---
>
> > **[W3]**: In Line 132, it says that “… whether a specific video was used during the instruction tuning state …”. What if the input video has been used in the pretraining stage? Will the method still work?
>
> **A3:** Thanks for this valuable comment. We believe that Vid-SME can be well applied in pretraining stage of VULLMs. However, **the current state of development of VULLMs is insufficient to support a broad and meaningful evaluation**. We elaborate on the reasons below.
>
> * Currently, **for most of the video understanding LLMs (VULLMs), the pretraining stage is typically performed on text-only corpora** (e.g., using a pretrained LLM as the language base model), and the incorporation of video data often begins during instruction tuning or modality alignment. Although there exits a few models (e.g., Qwen-2.5 VL) incorporate video data during the pretraining stage, those video data used in pretraining are not for the purpose of understanding, and model owners often do not disclose any information about the sources or composition of the pretraining corpus. As a result, membership inference in this context can only **serve as an internal auditing tool for the model owner**. For the external world, the lack of transparency makes it infeasible to construct matched member and non-member sets to conduct meaningful evaluations of privacy leakage. **This is also one of the key motivations behind our instruction-tuning three VULLMs**: allowing for controlled, fine-grained evaluation of membership leakage under different model conditions. There exits a lines of works focusing on the pretraining stage for **text-only LLMs** (e.g., the Pythia series, the model developers have publicly released the full set of pretraining corpora for research purpose). However, unfortunately, **no such dataset transparency currently exists for VULLMs**, and pretraining a VULLM from scratch is extremely resource-intensive and beyond our current computational capacity.
>
> * Furthermore, from a privacy perspective, pretraining datasets tend to consist of broad, publicly available or web-scale video sources, where **privacy concerns are less urgent**. In contrast, the task-specific datasets curated by developers in **the subsequent instruction tuning stage are more likely to include sensitive or proprietary information**. Since the creation of such datasets often involves considerable time and financial investment, they are particularly vulnerable to unauthorized usage or misappropriation by third parties, **making the instruction tuning phase both the most valuable and the most susceptible to membership inference attacks on VULLMs**. Therefore, we believe that focusing on the instruction tuning stage currently aligns better with practical deployment settings and realistic privacy threats.
>
> That said, **we fully agree with the reviewer** that performing membership inference on the pretraining stage is also an important and valuable direction, and therefore leave this as an important direction for future work.
>
> ---
>
> > **[W4]**:  Line 135, the notation of I(…) is not consistent with that in equation (1).
>
> **A4:** Thanks for catching this typo. We will correct it in the revision by replacing ⊕ with || in Line 135.
>
> ---
>
> Thanks again for the thoughtful and constructive feedback. If reviewer CyeB has any remaining concerns, we would definitely love to clarify.

---

> > ### Comment · Reviewer_CyeB · 2025-08-02
> >
> > I thank the authors for providing the detailed rebuttal, which mostly addressed my questions and concerns. I would therefore raise my rating accordingly.

---

> > > ### Author Response · Authors · 2025-08-02
> > > **Thanks for agreeing to raise the score**
> > >
> > > We are deeply encouraged to hear that reviewer CyeB’s questions and concerns have been addressed, and **we sincerely appreciate reviewer CyeB’s decision to further raise the rating** (from 4 to a higher score), this is truly a deeper and stronger affirmation of our work. Thanks again for reviewer CyeB's time and valuable input throughout the review process.

---

### Official Review · Reviewer_we3T · 2025-07-04

**Clarity:** 4
**Significance:** 3
**Originality:** 4
**Rating:** 5
**Confidence:** 4

**Summary:**

- The paper introduces a new Membership Inference Attack (MIA) method (Vid-SME) tailored for the temporal nature of video data for video understanding large language models (VULLMs)
- Existing MIA techniques designed for text or image data perform poorly when applied to video, largely due to their inability to account for the temporal dynamics and frame-dependent model behavior unique to video.
- To address this, the paper proposes Vid-SME, which uses Sharma-Mittal entropy to quantify model confidence under two conditions: when a video is shown in its natural order and when the frame sequence is reversed.
- The key insight is that VULLMs trained on a video are more confident (lower entropy) when frames are presented in their original order compared to reversed order.
- By measuring the entropy gap between these two settings and aggregating the smallest differences (via a K%-min pooling strategy), Vid-SME generates a membership score for each video.
- The paper also introduces a benchmark suite using three self-trained and two open-sourced VULLMs to evaluate the attack across diverse datasets, training strategies, and video types.
- Experiments demonstrate that Vid-SME significantly outperforms prior methods, particularly in low false positive rate (FPR) regimes, showing substantial gains in metrics like AUC and TPR@5%FPR.

**Questions:**

- Suggestion: Clarify the threat model. Could Vid-SME be adapted to black-box APIs?
- How are the Sharma–Mittal parameters q and r determined for each video? Are they chosen via optimization, heuristics, or fixed based on video metadata? Do these values generalize across datasets or need to be tuned for each model/video?
- Why is frame reversal specifically chosen as the transformation to reveal memorization? Have you considered or tested alternative temporal perturbations (e.g., frame shuffling, masking, partial reversal)?
- What is the computational overhead of Vid-SME per video inference?

**Ethical Concerns:**

["NO or VERY MINOR ethics concerns only"]

**Final Justification:**

The paper looks good to me, and I will continue with my original rating of accept.

**Limitations:**

Yes

**Quality:**

3

**Strengths And Weaknesses:**

-  VULLMs are increasingly trained on sensitive, real-world video data (e.g., surveillance, personal videos). The paper addresses an urgent and underexplored privacy risk - whether such models memorize and inadvertently expose their training data.
- The entropy-based scoring function using natural vs. reversed video sequences is a creative and well-motivated idea that leverages the temporal structure unique to videos—this is not a straightforward extension of prior MIA techniques.
- The adaptive use of this generalized entropy measure provides a novel and theoretically grounded mechanism to capture nuanced differences in model confidence across frames.
- The results are strong compared to the baselines and the paper goes beyond surface-level results by providing visualizations (e.g., Figure 2a–2b) that explain why the method works, showing how entropy differences become more distinct as video length increases.

Weaknesses:
- The attack appears to rely on logits or confidence scores from the model. It is unclear whether Vid-SME works when only hard labels (top-1 predictions) are available, which would be a more realistic setting for many deployed systems.
- While the paper emphasizes adaptive entropy parameterization, the method for choosing q/r values per video is underexplained. Is this based on optimization, empirical tuning, or predefined heuristics?
- The method requires computing entropy over possibly long sequences and for both natural and reversed orderings. There is no discussion of runtime or scalability with respect to video length or number of frames.

---

> ### Author Rebuttal · Authors · 2025-07-28
>
> It’s our great honor to receive Reviewer we3T’s thoughtful comments and strong support for our work. We would like to address the concerns as below.
>
> ---
>
> > **[Q1, W1]**: Suggestion for clarify the threat model.
>
> **A1:** Thanks for the valuable suggestion. Vid-SME is designed for the gray-box scenario, which is a commonly adopted setting in prior works (e.g., Rényi entropy and Min-K%). This is attributed to its broad applicability in real-world scenarios:
>
> * For example, **closed-source model APIs such as OpenAI's GPT-4 API allow users to retrieve the top-five per-token probabilities**, which also can be seen as a gray-box scenario in our context and the adversaries can exploit this information to conduct attacks. We follow this setting (where only the top-five per-token probabilities are accessible) and conduct an experiment on Video-XL-CinePile-7B with the number of frames set to 8. As shown in the table below, Vid-SME remains reliable in performing membership inference under this partial-knowledge scenario.
>
> | Method        | Variant   | AUC   | Acc.  | TPR@5% FPR |
> |---------------|-----------|-------|-------|-------------|
> | Vid-SME    | Mean   | 0.775 | 0.703 | 0.305 |
> || Min_60%   | 0.657 | 0.628 | 0.229 |
>
> * Also, such attacks are not only relevant for external adversaries, but also provide a **practical auditing tool** for model developers and owners to monitor potential privacy risks in their deployed models.
>
> * Moreover, the gray-box setup represents a **conservative risk lower-bound**. If an attack succeeds under the gray-box setting, it indicates that the system may already be operating at the boundary of its privacy guarantees.
>
>
>
> We sincerely hope that our responses can alleviate this concern and we will further clarify our threat model and its practical motivations in the revision.
>
> ---
>
> > **[Q2, W2]**: How are the Sharma–Mittal parameters q and r determined for each video? Do these values generalize across datasets or need to be tuned for each model/video?
>
> **A2:** Thanks for the valuable comment. When designing the q/r value, we took into account the following key considerations:
>
> (1). Whether the computation remains naturally consistent under different video conditions (e.g., varying frame numbers);
>
> (2). Whether the q/r values are compatible with the Sharma-Mittal Entropy calculation;
>
> (3). Whether the q/r values can adequately capture the temporal nature and complex inter-frame variations of the video.
>
> In response to (1), once a video's sampled frames are determined, its q/r values become fixed, i.e., as you noted, the q/r values are **based on video metadata**. This means the **q/r values are model-agnostic (do not need to be tuned per model)** and those model-specific informations are captured through the subsequent entropy computation. The q/r values only need to be re-computed when the frame condition changes (e.g., from 8 to 16 frames). In response to (2) and (3), we specifically designed the q/r values to reflect video-specific temporal dynamics and inter-frame variations through motion complexity and illumination variation. We will further clarify these design choices and details in the revision.
>
> ---
>
> > **[Q3]**: Why is frame reversal specifically chosen as the transformation to reveal memorization? Have you considered or tested alternative temporal perturbations (e.g., frame shuffling, masking, partial reversal)?
>
> **A3:** Thanks for the insightful and thought-provoking comment. We would like to share our perspective on this point from two perspectives:
>
> * First, we agree that applying transformations such as frame shuffling, masking, or partial reversal is a meaningful direction worth exploring. However, such approaches naturally introduce **additional hyperparameters**, such as which frames to shuffle, which regions to mask, and which segments to reverse. This leads to a much broader design space, and hyperparameter choices may vary across different videos. While some carefully crafted strategies may indeed further improve the attack performance, our choice of full-frame reversal is a **more intuitive and lightweight solution**, requiring no tuning while already yielding strong results.
>
> * Second, as an analogy to injecting noise within each frame,  applying masking can be seen as **a process of frame corrupting**. Some prior work (e.g., Rényi entropy) as well as our own experiments (see Figure 4(c) and Lines 374-379) have taken this into consideration by **evaluating the robustness of the method** under frame corruption settings.
>
> Overall, we truly appreciate the reviewer’s suggestion on exploring alternative ways to incorporate video-specific properties to reveal memorization in VULLMs. We see this as a valuable and exciting direction and plan to pursue it further in future work.
>
> ---
>
> > **[Q4, W3]**: What is the computational overhead of Vid-SME per video inference?
>
> **A4:** Thanks for raising this point. In the table below, we provide a comparison of inference time per video and attack performance between the best-performing baseline Rényi entropy and Vid-SME. The experiments are conducted using the Video-XL-CinePile-7B model with frame numbers set to 8 and 16, with both methods using the variant of Max_90%/Min_90% and  α in Rényi entropy is set to 2 (note that the variant used and the α value in Rényi entropy do not affect the inference time).
>
> _\# Frames = 8:_
>
> | Method        | Variant   | AUC   | Acc.  | TPR@5% FPR | Time per Video (s) |
> |---------------|-----------|-------|-------|-------------|---------------------|
> | Rényi | Max_90%   | 0.559 | 0.585 | 0.004       |        6.11             |
> | Vid-SME    | Min_90%   | 0.806 | 0.760 | 0.289 |       9.45              |
>
> _\# Frames = 16:_
>
> | Method        | Variant   | AUC   | Acc.  | TPR@5% FPR | Time per Video (s) |
> |---------------|-----------|-------|-------|-------------|---------------------|
> | Rényi  | Max_90%   | 0.661 | 0.655 | 0.004       |        11.67             |
> | Vid-SME    | Min_90%   | 0.830 | 0.765 | 0.305 |       16.76              |
>
> * Compared to Rényi entropy, the additional computational cost of Vid-SME lies in **the computation of the adaptive q/r values**. When these video-specific values are not computed, the inference-time cost of Vid-SME is nearly identical to that of Rényi entropy, as **both require two forward queries to the model**.
>
> * Also, as the number of frames increases, Vid-SME demonstrates **good scalability in both inference time and performance improvement**.
>
> We sincerely hope that our responses adequately address the point and we will include the additional comparisons on time consumption in the revision.
>
> ---
>
> Thanks again for the thoughtful and constructive feedback. We would definitely love to further interact with the reviewer if there are any further questions.

---

### Decision · Program_Chairs · 2025-09-17

**Decision:**

Accept (poster)

**Comment:**

This paper introduces the first membership inference method specifically designed for video data utilized in video understanding large language models (LLMs), termed Vid-SME. This method harnesses the confidence of model outputs and incorporates adaptive parameterization to calculate Sharm-Mittal entropy (SME) for video inputs. Experiments conducted on a variety of self-trained and open-sourced video understanding LLMs (VULLMs) validate the efficacy of Vid-SME. All reviews acknowledge the significance of the generalized entropy measure. The authors' rebuttal has effectively addressed the majority of concerns raised by the reviewers. Overall, I recommend the acceptance of this submission. Additionally, I anticipate that the authors will integrate the new results and suggested modifications from the rebuttal phase into the final version.